# RPAP3 C-Terminal Domain: A Conserved Domain for the Assembly of R2TP Co-Chaperone Complexes

**DOI:** 10.3390/cells9051139

**Published:** 2020-05-06

**Authors:** Carlos F. Rodríguez, Oscar Llorca

**Affiliations:** Structural Biology Programme, Spanish National Cancer Research Centre (CNIO), Melchor Fernández Almagro 3, 28029 Madrid, Spain; cfrodriguez@cnio.es

**Keywords:** RUVBL1, RUVBL2, RPAP3, PIH1D1, R2TP, HSP90 co-chaperone, TPR domain, URI1 prefoldin complex

## Abstract

The Rvb1-Rvb2-Tah1-Pih1 (R2TP) complex is a co-chaperone complex that works together with HSP90 in the activation and assembly of several macromolecular complexes, including RNA polymerase II (Pol II) and complexes of the phosphatidylinositol-3-kinase-like family of kinases (PIKKs), such as mTORC1 and ATR/ATRIP. R2TP is made of four subunits: RuvB-like protein 1 (RUVBL1) and RuvB-like 2 (RUVBL2) AAA-type ATPases, RNA polymerase II-associated protein 3 (RPAP3), and the Protein interacting with Hsp90 1 (PIH1) domain-containing protein 1 (PIH1D1). R2TP associates with other proteins as part of a complex co-chaperone machinery involved in the assembly and maturation of a growing list of macromolecular complexes. Recent progress in the structural characterization of R2TP has revealed an alpha-helical domain at the C-terminus of RPAP3 that is essential to bring the RUVBL1 and RUVBL2 ATPases to R2TP. The RPAP3 C-terminal domain interacts directly with RUVBL2 and it is also known as RUVBL2-binding domain (RBD). Several human proteins contain a region homologous to the RPAP3 C-terminal domain, and some are capable of assembling R2TP-like complexes, which could have specialized functions. Only the RUVBL1-RUVBL2 ATPase complex and a protein containing an RPAP3 C-terminal-like domain are found in all R2TP and R2TP-like complexes. Therefore, the RPAP3 C-terminal domain is one of few components essential for the formation of all R2TP and R2TP-like co-chaperone complexes.

## 1. R2TP: A Versatile and Complex Co-Chaperone Machinery

Heat shock protein 90 (HSP90) is a molecular chaperone that assists the maturation, stability, and assembly of a number of cellular proteins and macromolecular complexes. HSP90 works with the help of co-chaperones, nonclient partners that interact with the chaperone and contribute to its function. Some co-chaperones contain tetratricopeptide repeat (TPR) domains that bind to EEVD motifs located at the C-terminus of each HSP90 monomer and help recruit HSP90 to specific clients [1]. Crystal structures of TPR domains cocrystallized with small peptides containing the EEVD C-terminal motifs reveal that each HSP90 C-terminal tail binds the groove formed by five of the α-helices of a TPR domain [2,3].

R2TP is an HSP90 co-chaperone made of four different subunits (see later), one of which, RPAP3, is a protein containing two TPR domains that interact with HSP90 and Heat shock protein 70 (HSP70) [2,3,4]. The R2TP complex was first described in *Saccharomyces cerevisiae* and it was later found in mammalian cells [4,5,6]. The R2TP complex works together with HSP90 in the stability and assembly of Pol II [7], small nucleolar ribonucleoproteins (snoRNPs) [8], and complexes of the PIKK family of kinases, including mTOR complexes [9,10]. Evidence suggests that R2TP also participates in the assembly and activation of other large macromolecular assemblies, such as U5 snRNP, the TSC1-TSC2 tuberous sclerosis complex, and possibly several others [6,11,12]. Current models propose that R2TP brings together clients and HSP90, facilitating the action of the chaperone on the client.

R2TP is made of four subunits: RUVBL1, RUVBL2, RPAP3, and PIH1D1 (Figure 1) [4,5]. RUVBL1 and RUVBL2 are closely related AAA-ATPases that assemble RUVBL1-RUVBL2 hetero-hexameric complexes with alternating subunits. RPAP3 was identified during a systematic analysis of the protein interaction network for Pol II [13]. Several proteins were associated with Pol II and named RNA polymerase II (RNAP II)-associated proteins (RPAPs). RPAP3 contains two TPR domains capable of recruiting HSP90 and HSP70 chaperones [2,3]. On the other hand, PIH1D1 contains a PIH domain, a phosphopeptide-binding module recognizing a specific acidic motif after phosphorylation by casein kinase 2 (CK2) [3,10,14]. This consensus sequence has been found in several proteins that interact with R2TP, such as MRE11, a potential client of R2TP [15], and TELO2, a protein required for the assembly of the kinases of the PIKK family (see later) [3,9,14].

A fully functional R2TP-based chaperone pathway requires the participation of additional proteins (Figure 1). R2TP associates with several other proteins, which may be important for some of its functions [11]. R2TP interacts with the URI1 prefoldin complex, which is composed by prefoldin-like proteins PFDN2, PFDN6, URI1, UXT, PDRG1, and the recently discovered ASDURF [16]. The drug α-amanitin induces Poll II disassembly and the association of several subunits of Pol II with R2TP and the URI1 prefoldin complex [7]. This has been interpreted as an indication that the complex between R2TP and URI1 prefoldin complex participates in the maturation of Pol II. The WDR92/Monad protein also associates with R2TP and the URI1 Prefoldin complex and, this interaction is important in the regulation of ciliary dynein preassembly and axonemal dynein heavy chain stability in the cytoplasm [17,18,19].

Some of the proteins that interact with R2TP work as adaptors connecting the R2TP co-chaperone machinery to specific clients (Figure 1). The complex formed by TELO2, TTI1, and TTI2, known as the TTT complex, helps bring the PIKK family of kinases to R2TP [9]. The TTT complex forms a complex with these kinases, whereas the PIH domain in PIH1D1 recognizes the phosphorylated version of the PIH consensus motif in TELO2 [10,14]. In addition, the protein WAC interacts with R2TP and the TTT complex and regulates the activation of mTORC1 complexes by yet unknown mechanisms [20]. Zinc finger HIT domain-containing protein 2 (ZNHIT2) and ecdysoneless cell cycle regulator (ECD) have been proposed to function as R2TP adaptors in the maturation of U5 snRNP [12,21]. ZNHIT2 belongs to a family of zinc finger histidine triad (HIT) domain-containing proteins with several members forming complexes with RUVBL1-RUVBL2 [22]. ZNHIT1 and ZNHIT4 interact with RUVBL1-RUVBL2 as part of the chromatin remodeling complexes SRCAP [23] and INO80 [24], respectively, whereas ZNHIT3 and ZNHIT6 collaborate with R2TP and HSP90 in the assembly of box C/D snoRNPs [22]. During the box C/D snoRNP assembly pathway, several intermediate complexes are formed, including a complex containing assembly factors NUFIP1, ZNHIT3, and ZNHIT6 associated to RUVBL1-RUVBL2. NUFIP1, ZNHIT3, and R2TP also participate in the biogenesis of U4 snRNP [25,26]. Thus, the R2TP-based chaperone system could be extraordinarily versatile. Some subsets of adaptors would be involved in the assistance to certain clients and not others, and the participation of some proteins and not others could serve to regulate specific activities and function.

Recently, Houry, Bertrand, and Coulombe proposed renaming the chaperone machinery composed by R2TP, WDR92, RPB5 (a component of all eukaryotic RNA polymerases), and the URI1 prefoldin complex as PAQosome (particle for arrangement of quaternary structure) to better reflect the function of this co-chaperone machinery in the acquisition of the quaternary structure of a growing number of large macromolecular complexes (Figure 1) [11]. Some of the subunits of the PAQosome complex have homologous proteins and there is evidence supporting that these proteins can assemble alternative forms of the R2TP complex (see later) and alternative PAQosomes [6,27]. It is important to indicate that although the URI1 prefoldin complex associates with R2TP, current evidence only supports a potential role of this association during the assembly of Pol II whereas most of the functions currently identified for R2TP appear to be independent of the URI1 prefoldin complex.

## 2. RUVBL1 and RUVBL2, Two ATPases at the Core of the R2TP

Of all the components currently identified as part of the R2TP-based co-chaperone system, only the RUVBL1 and RUVBL2 ATPases are consistently found to be essential in all the activities described, suggesting that these are non-interchangeable components of the system. RUVBL1 and RUVBL2 are two closely related AAA-type ATPases that form hetero-hexameric rings where three copies of RUVBL1 and three copies of RUVBL2 subunits alternate [4] (Figure 2A). This hexameric RUVBL1-RUVBL2 complex is an essential component not only of R2TP but also of other macromolecular complexes, such as the INO80 and SWR1 chromatin-remodelers [28,29]. In these complexes, the ATPase hexameric ring serves as a scaffold for the interaction of other proteins in the complex.

The atomic structure of RUVBL1-RUVBL2 has been studied by X-ray crystallography in several species: *Saccharomyces* cerevisiae, *Chaeetomiun thermophilim*, and *Homo sapiens* [30,31,32,33]. The complex displays a very similar architecture in all the species analyzed, and each subunit consists of three domains. Domain I (DI) and domain III (DIII) form the core of the protein that contains the nucleotide binding domains of RUVBL1 and RUVBL2 and is responsible for the oligomerization (Figure 2A). Domain II (DII) is flexible, projects from each subunit of the hexamer, and it is involved in the interaction with other proteins. DII domains contain an internal segment in proximity to the hexameric core and an external region made of an oligonucleotide-binding (OB) fold domain. Several clients and adaptors have been found to interact with both elements of the DII domains [29,34]. In all the RUVBL1-RUVBL2 complexes yet analyzed, it is the DII-face of the hexameric ring that is mostly involved in protein–protein interactions [4,28,29]. RUVBL1 and RUVBL2 can also form dodecameric complexes made of two stacked back-back hexameric rings interacting by their DII domains. Although some functions have been proposed for dodecameric complexes, available evidence suggests that these double-ring complexes could be a storage form before the DII face of the ring is used as a scaffold for protein–protein interactions [35,36].

## 3. Organization of the R2TP Complex

In 2017, two cryo-EM structures of yeast R2TP (Rvb1p-Rvb2p-Tah1p-Pih1p) reconstructed at medium resolution revealed that Rvb1p-Rvb2p formed a hexameric ring that scaffolds a Tah1p (yeast RPAP3 homologue) in complex with Pih1p (yeast PIH1D1 homologue) [38,39]. Pih1p and Tah1p were located at the DII face of the ATPase ring and as a consequence, the client recruitment PIH domain of Pih1p and the TPR domain in Tah1 necessary for HSP90 recruitment were in proximity.

A significant advance in our structural understanding of human R2TP was achieved in 2018 [37] (Figure 2B,C). Compared to yeast Tah1p, RPAP3 is a much larger (75 kDa) multi-domain protein containing two TPR domains at the N-terminal half of the protein, each of which has been shown to bind HSP90, although with different affinity [2,3] (Figure 3). A C-terminal region in RPAP3 corresponding to residues 546–665 was annotated as a protein domain (pfam13877) and which we name RPAP3 C-terminal domain and RPAP3-Cter, hereafter. The region connecting this C-terminal region and the second TPR domain is predicted as disordered or low complexity. Yeast Tah1p is a smaller 12 kDa protein that lacks the RPAP3 C-terminal domain and contains one TPR domain [3] (Figure 3).

The cryo-EM structure of human R2TP revealed some similarities to the yeast complex. In human R2TP, PIH1D1 and most of RPAP3 are located in the DII face of the RUVBL1-RUVBL2 ring as in the yeast complex, and this allows R2TP to recruit clients and HSP90 in proximity, facilitating the action of the chaperone on its clients. In addition to the structural similarities with yeast R2TP, the human complex revealed unique structural features. RPAP3 is the only protein that interacts with all other components of the human R2TP complex, and it is therefore essential for R2TP assembly, a function performed by Pih1p in yeast. In addition, the N-terminal TPR containing half in RPAP3 is very flexible compared to the restricted flexibility of the components of yeast R2TP (Figure 2B,C). Differences are also found in the stoichiometry of the human R2TP complex, which contains up to three RPAP3 molecules for each RUVBL1-RUVBL2 ring (see below) (Figure 2A).

The structural characterization of the R2TP complex is completed by the characterization of the interaction of PIH1D1 with the DII domain of an RUVBL2 subunit and how this interaction affects the structure of RUVBL1-RUVBL2 [34] and by a detailed structural characterization of the TPR domains of RPAP3 using X-ray crystallography and NMR [2,3]. The crystal structures of both TPR domains of RPAP3 were solved in complex with the HSP90 C-terminal MEEVD motif. The TPR domains comprise seven α-helices and the first five conform the structural motif for binding to the HSP90 C-terminal tail. Interaction and mutations studies based on these structures suggested that the N-terminal TPR domain (TPR1) binds HSP90 with higher affinity than the second TPR domain (TPR2) [3]. Another study determined the structure and affinities of both TPR domains for HSP90 and HSP70 C-terminal peptides using NMR and short and long versions of the C-terminal tails [2]. Interestingly, the affinity of the longer version of the HSP90 C-terminal tails for TRP2 was around 20 times higher than for TRP1. The length of the C-terminal peptide was an important determinant of the affinity because residues before the MEEVD motif were essential for strong binding to TPR2. Interestingly, both TPR domains can bind HSP70 C-terminal peptides, and SILAC experiments using TPR1 as bait revealed that the association of HSP70 to TPR1 occurred in cells [2]. How HSP70 collaborates with HSP90 and R2TP in some of their biological activities remains to be elucidated.

## 4. Structure of the RPAP3 C-Terminal Domain

The most noticeable difference between yeast and human R2TP complexes is the role of the RPAP3 C-terminal domain in the assembly of the full complex [34,37]. Cryo-EM shows that the RPAP3 C-terminal domain interacts with the ATPase face of each RUVBL2 subunit (Figure 2A). This is, so far, the only protein domain found that binds to the ATPase face of the RUVBL1-RUVBL2 ring, opening the possibility that other proteins could use a similar surface to form a complex with these ATPases. Each RPAP3 C-terminal domain binds to one RUVBL2 subunit in the ring, and accordingly the domain was also named the RUVBL2-binding domain (RBD). Although there are three RUVBL2 subunits per RUVBL1-RUVBL2 hexamer, R2TP complexes reconstituted in vitro can contain one, two, or three RPAP3 per RUVBL1-RUVBL2 ring. Until now, the functional significance of these various forms remains unknown.

The atomic structure of the RPAP3 C-terminal domain has been determined as part of R2TP using cryo-EM [37] (Figure 4A) as well an individual domain using NMR [27] (Figure 4B). Cryo-EM of the human R2TP complex revealed that the RPAP3 C-terminal domain is a bundle of eight α-helices interacting with RUVBL2 (Figure 4A). Cryo-EM, cross-linked mass spectrometry, and NMR suggest that the RPAP3 C-terminal domain is preceded by a long unstructured region that connects the TPR and RPAP3 C-terminal domains through the outer rim of the RUVBL1-RUVBL2 ring [27,37] (Figure 2C). The NMR structure of the RPAP3 C-terminal domain mostly agrees with the conformation of the domain described by cryo-EM (Figure 4B). Recently, we have provided a detailed comparison and analysis of the two structures [40]. Observed minor discrepancies could be due to differences in the conformation of an isolated domain in solution, and when the same domain is in the context of the complete RPAP3 protein in complex with the rest of the components of R2TP.

## 5. The RPAP3 C-Terminal Domain Is Required to Assemble the R2TP Complex

The cryo-EM structure of the R2TP complex shows regions of the RPAP3 C-terminal domain facing RUVBL2, and some residues that can be important for their interaction [37] (Figure 4C). Helices 1 and 6 of the RPAP3 C-terminal domain that contact RUVBL2 are 100% identical in a number of species, suggesting their relevance in the formation of the complex. Maurizy and colleagues have shown that an alanine double mutant of residues Arg623 and Met626 that are located next to each other in helix 6 abolished the interaction between the RPAP3 C-terminal domain and RUVBL1-RUVBL2 [27] (Figure 4C). Arg623 interacts with Asp435 in RUVBL2 and this interaction is stable in molecular dynamics simulations [40]. In addition, the double mutant Phe630-Ser632 also disrupted the interaction of the RPAP3 C-terminal domain and RUVBL1-RUVBL2 [27] (Figure 4C).

The RPAP3 C-terminal domain performs at least two essential roles. On the one hand, it assembles R2TP by bringing together RPAP3 (or the RPAP3-PIH1D1 complex) and the RUVBL1-RUVBL2 ATPase complex. In vitro mapping experiments have shown that the interaction between the RPAP3 C-terminal domain and RUVBL2 is essential to hold the R2TP complex together [37]. PIH1D1 forms a complex with RPAP3 and it also interacts with RUVBL1-RUVBL2, but after truncation of the RPAP3 C-terminal domain, neither RPAP3 nor PIH1D1 can assemble in a complex with RUVBL1-RUVBL2 [37]. On the other hand, the RPAP3 C-terminal domain provides a tight anchor for RPAP3 at the ATPase face of the RUVBL1-RUVBL2 hetero-hexameric ring, thus permitting the flexibility of the RPAP3 N-terminal regions at the DII face of the ring (Figure 2B,C). We proposed that the flexibility of these regions could have a functional significance in the binding of HSP90 to its clients in the context of the R2TP complex [37]. A rigid structure of the R2TP complex would probably be incompatible with the interaction of HSP90 with clients of different composition and structure, such as mTOR, ATR, and Pol II. However, the strong interaction provided by the RPAP3 C-terminal domain frees the conformation of the HSP90-recruiting regions of RPAP3, which can adapt to various clients. The flexibility of the N-terminal regions of R2TP was clearly visualized in cryo-EM images of the R2TP complex (Figure 2B). To determine whether the observed flexibility has a functional role, further understanding of the structures of larger intermediates containing R2TP, clients, chaperones, and adaptors, and in general additional knowledge on how the R2TP-based chaperone system works is required.

## 6. RPAP3 C-Terminal-Like Domains and the Assembly of R2TP-Like Complexes

RPAP3 exists in two isoforms, with isoform 2 being shorter as it lacks few residues essential for the interaction of RPAP3 with PIH1D1 [27]. In addition, two other human proteins, SPAG1 and CCDC103, contain domains with homology to the RPAP3 C-terminal domain [27] (Figure 3). SPAG1 participates in the assembly of dynein arms and contains three rather than two TPR domains and a RPAP3 C-terminal-like domain at the C-terminus [41,42]. CCDC103 is a smaller protein comprising an RPAP3 C-terminal-like domain located at the center rather than at the C-terminal end of the protein and flanked by coiled coils domains. CCDC103 is also involved in the assembly of the outer dynein arms on ciliary axonemes [43]. In vitro assays revealed that CCDC103 self-assemble into oligomers and a short disordered region (residues 68–94) adjacent to the RPAP3 C-terminal-like domain is sufficient to mediate dimer formation [44]. Residues 93 to 242 that include the RPAP3 C-terminal domain of CCDC103 (residues 96–189) also revealed oligomeric properties in vitro. Several deleterious mutations in CCDC103 are found in primary ciliary dyskinesia (PCD) and the related *situs inversus* disease, suggesting a critical role of CCDC103 in the assembly of dynein arms on ciliary axonemes [45]. A mutation found in PCD and situs inversus disease is located specifically in the CCDC103 RPAP3 C-terminal domain (His154Pro), yielding an unstable protein in vitro [46].

The canonical R2TP complex is made of RUVBL1-RUVBL2 hetero-hexamer in complex with RPAP3 and PIH1D1 [37], but the existence of PIH1D1-like and RPAP3-like proteins suggests that alternative forms of the R2TP complexes are possible. PIH1D1 contains a PIH-domain, which is present in other proteins, PIH1D2, PIH1D3, and DNAAF2. The RPAP3 C-terminal-like domains of RPAP3, SPAG1, and CCDC103 have roughly a 25% sequence identity [27], and we modeled their structure using the structure of RPAP3 as a reference [40]. These predictions suggested that the RPAP3 C-terminal-like domains of SPAG1 and CCDC103 could have a similar structure to RPAP3 and the potential to bind to RUVBL1-RUVBL2 [40]. Using pairwise LUMIER co-IP assays as well as SILAC proteomic analysis using the GFP-fused version of PIH1D2, PIH1D3, DNAAF2, and CCDC103, Maurizy and colleagues demonstrated that SPAG1 interacts with RUVBL1-RUVBL2 and they identified the existence of two complexes containing SPAG1 and RUVBL1-RUVBL2, bound to either PIH1D2 or DNAAF2 [27] (Figure 5). The complex comprising SPAG1, PIH1D2, and RUVBL1-RUVBL2 likely shares an organization similar to R2TP, but where RPAP3 and PIH1D1 have been substituted by SPAG1 and PIH1D2, and the complex was named R2SP. Similarly, the complex comprising SPAG1, DNAAF2, and RUVBL1-RUVBL2 was named R2SD but remains hypothetical because it was not detected in their proteomic experiments. These two complexes would have a role in the assembly of the outer and inner dynein arms [41]. Noteworthy, an interaction between the RPAP3 C-terminal-like domain of SPAG1 and RUVBL1-RUVBL2 was not observed, suggesting that some binding determinants outside the domain are required for the assembly of the complex. Finally, Maurizy and colleagues identified a third complex, named R2T, made of isoform 2 of RPAP3, and thus unable to bind PIH1D1 (Figure 5). In their experiments, they did not detect any partner of CCDC103, not even RUVBL1-RUVBL2, remaining unclear if this protein can bind RUVBL1-RUVBL2.

In summary, RPAP3 C-terminal-like domains allow the interaction between an RPAP3-like protein and the RUVBL1-RUVBL2 ATPases forming several distinct complexes, collectively named as R2TP-like [27] (Figure 5). The functions of these R2TP-like complexes have not been studied in detail yet, but it has been suggested that some of these could have tissue-specific roles and specificity towards some clients.

## 7. Conclusions

Pol II, U5 snRNP, and mTORC1 complexes are among several large macromolecular complexes that require the action of the R2TP-based chaperone system for their assembly, activation, or maturation. This chaperone system is versatile, with not all the components working for each client, and with several RPAP3-like and several PIH1D1-like proteins implicated. The interaction of the RUVBL1-RUVBL2 ATPases with a protein containing an RPAP3 C-terminal-like domain is the only element apparently conserved in all versions of the R2TP-based chaperone. Further progress needs to be made to identify all the proteins involved in the assistance to each client and their specific role. The role of ATP binding and ATP hydrolysis by RUVBL1 and RUVBL2 in the R2TP complex is still mysterious and this needs further investigation. Solving the structures of the URI1 prefoldin complex and its interaction with R2TP, as well as the structures of complexes between R2TP and adaptors, clients, and chaperones will be important to fully understand how R2TP functions.

## Figures and Tables

**Figure 1 cells-09-01139-f001:**
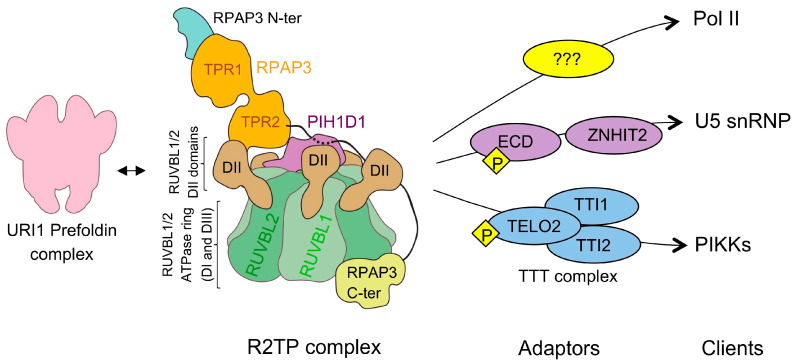
The R2TP-based chaperone system. The cartoon represents the R2TP complex and its constituent proteins. The RUVBL1-RUVBL2 complex (labeled as RUVBL1/2 in the figure for simplicity) forms a hexameric ring that contains the ATPase domains and made of domains I (DI) and III (DIII). Domain II (DII) protrudes from each subunit and it can interact with other proteins. In the cartoon, the RUVBL1-RUVBL2 complex is seen from its side. The maturation of some clients requires the action of adaptors that have been proposed to link R2TP to its clients. Some adaptors contain a consensus motif recognized by PIH1D1 after CK2 phosphorylation, and in the figure, a “P” represents the phosphorylated version of this motif. The URI1 prefoldin complex associates with R2TP, and this interaction could be implicated in the assembly of Pol II. It is unknown if the functions of R2TP in the assembly of Pol II require specific adaptor proteins, and this is indicated as several question marks.

**Figure 2 cells-09-01139-f002:**
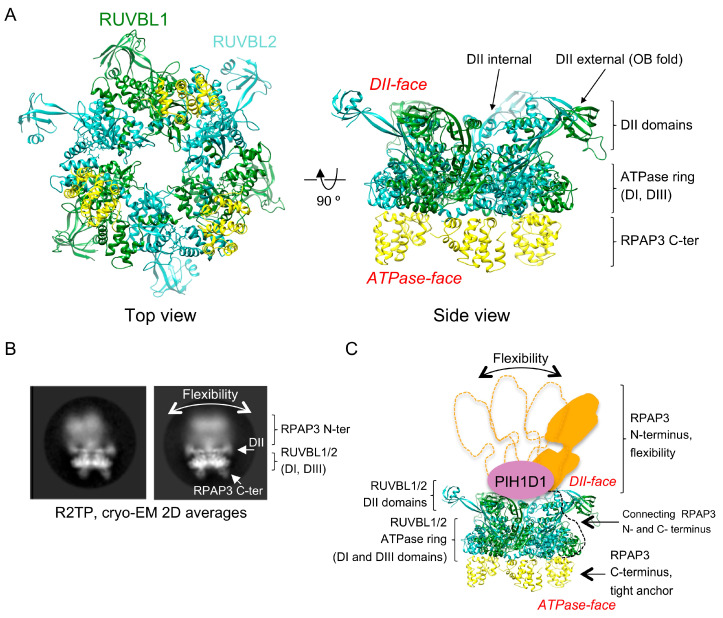
Structure of RUVBL1-RUVBL2 AAA-ATPases and the R2TP complex. (**A**) Top and side view of the atomic structure of the core of the human R2TP complex comprising one RUVBL1-RUVBL2 hexameric ring bound to three copies of the RPAP3 C-terminal domain [37]. Subunits, domains, and regions are indicated. (**B**) Two representative 2-D averages of cryo-EM images obtained for the R2TP complex [37]. Domains of RUVBL1-RUVBL2 and RPAP3 are indicated. (**C**) Cartoon representing the location of PIH1D1 and RPAP3 N-terminal end at the DII-face of the RUVBL1-RUVBL2 ring, and the flexibility of RPAP3 is indicated.

**Figure 3 cells-09-01139-f003:**
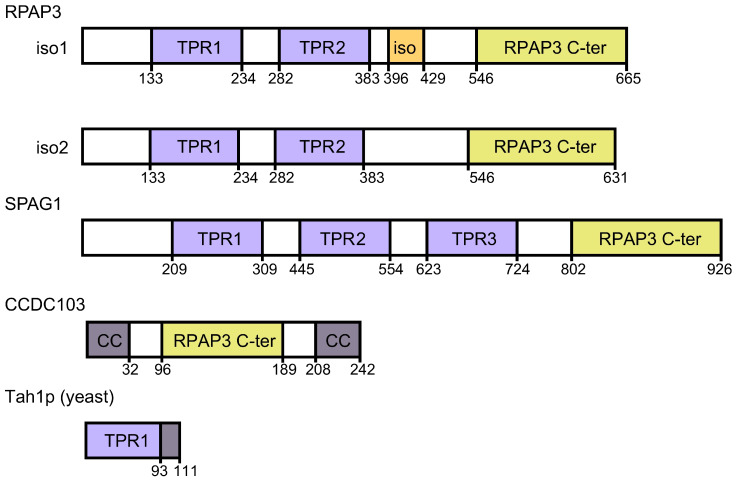
RPAP3 and homologous proteins. List of proteins homologous to human RPAP3. Homologous domains to the C-terminal domain of RPAP3 are labeled as RPAP3 C-ter. Iso1 and iso2 correspond to the two isoforms of RPAP3. The region in RPAP3 involved in binding to PIH1D1, present in isoform 1 but not in isoform 2, is labeled as “iso” (orange).

**Figure 4 cells-09-01139-f004:**
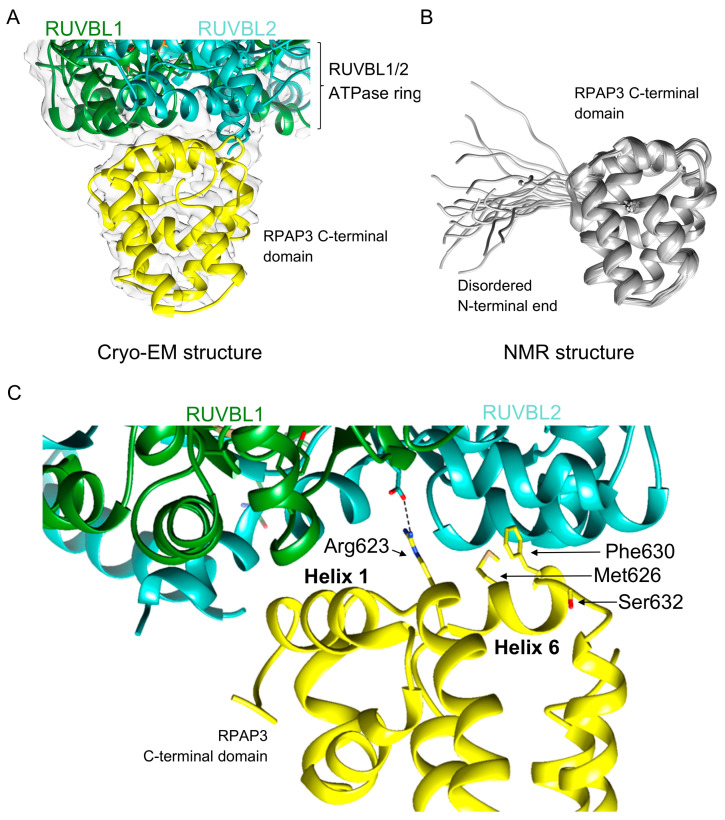
Structure of the RPAP3 C-terminal domain. (**A**) One view of the structure of the RPAP3 C-terminal domain obtained using cryo-EM [37]. The structure shows the interaction with the RUVBL1-RUVBL2 complex. The experimental cryo-EM map is represented as a transparent density. (**B**) NMR coordinates of the RPAP3 C-terminal domain [27]. (**C**) Close-up of the region of contact between the RPAP3 C-terminal domain and RUVBL2 observed by cryo-EM [37], indicating the residues mutated by Maurizy and colleagues [27].

**Figure 5 cells-09-01139-f005:**
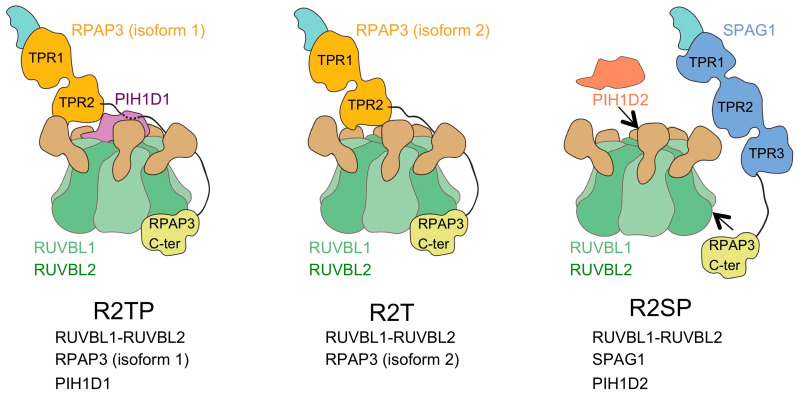
Model for R2TP-like complexes. R2TP and several R2TP-like complexes have been identified [27]. For R2SP, the possible interactions between the components in the complex are only indicated, since the structure and organization of R2SP has not been characterized in detail.

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
