# Peer review of "RPAP3 C-Terminal Domain: A Conserved Domain for the Assembly of R2TP Co-Chaperone Complexes"

_cells, 2020, doi:10.3390/cells9051139_

Round 1

Reviewer 1 Report

Review: RPAP3 C-terminal domain, a conserved domain for the assembly of R2TP co-chaperone complexes

The review stresses an important point: the RPAP3 C-terminal may be essential for the assembly of R2TP and R2TP-like complexes. The authors have done a nice job reviewing the current literature on R2TP structures, R2TP-like complexes, the RPAP3 C-terminal domain, and all known proteins containing the RPAP3 C-terminal domain (Sections 3-6). The review would benefit from a more in-depth analysis of the literature regarding the functions of adaptor proteins, namely NUFIP, and perhaps WAC, ZNHIT3, and ZNHIT6 (Section 1). The authors also need to be more precise in their use of the nomenclature (Section 1). The PAQosome does not include adaptor proteins or proteins that comprise R2TP-like complexes (ie., SPAG1 and PIH1D2). Also, there are several instances where the authors imply that the URI1 prefoldin complex is involved in functions that have been linked exclusively to R2TP. We know that the URI1 prefoldin complex associates with R2TP and may be involved in RNA Polymerase II assembly, but that’s it.

This draft needs to undergo editing for language as is suffers from constant use of the passive voice, wordiness, and grammatical errors.

Below are more specific comments:

Abstract

  • Line 13 – Hyphen missing in “RNA Polymerase II associated protein 3”
  • Line 19 – “complexes that could have” should be “complexes, which could have”

Section 1

  • Line 27 – HSP90 abbreviation not mentioned
  • Line 32 – There needs to be an additional reference. The reference provided does not have experiments showing where TPR-domain containing co-chaperones are binding on Hsp90; it only shows that they bind.
  • Lines 33,34 – Very awkward and wordy sentence. Perhaps mention the PAQosome here instead.
  • Line 36 – Pol II abbreviation introduced a second time
  • Line 37 – PIKK abbreviation introduced a second time
  • Line 38 – “suggest” should be “suggests”
  • Lines 44, 45 – I suggest saying “R2TP is made of four subunits: RUVBL1, RUVBL2, RPAP3 and PIH1D1” No need to mention “AAA-ATPase” if the following sentence mentions it.
  • Lines 46-48 – Another very awkward sentence that should be re-cast.
  • Line 52 – “or in TELO2” should be “and TELO2”
  • Line 52 – “TELO2, which” should be “TELO2 that”
  • Line 52 – Proposed? It’s been shown through experiments.
  • Figure 1:
    • “URI1 prefoldin” – I think they meant “URI1 prefoldin complex”
    • Brown shapes are not labeled or indicated in the figure legend
    • znHIT2 should be “ZNHIT2”
  • Line 56 – “associates to” should be “associates with”
  • Line 57 – “factors” – find a better word. Perhaps say, “the action of “adaptors” that have been proposed to link R2TP to its clients.”
  • Line 61 – fix sentence
    • “R2TP associates in cells with” should be “R2TP associates with”
    • “Several other factors and subunits” should simply say, “several other proteins”
      • Subunits of what?
    • “, which seem to be” should be, “which may be”
      • Prefoldin has not been shown to be important, at least not yet
    • Line 62 – Why “so-called?”
    • Line 64 – “ASDRURF” should be “ASDURF”
    • Line 65 – “RNA polymerase” – there are three RNA polymerases and RPB5 is a subunit of each one
    • Line 65 – “WDR92/Monad protein, and this might be an indication that the R2TP/URI1 Prefoldin like complex participates together with HSP90 …”
      • Is the author referring to R2TP’s association with WDR92? Because WDR92 was not shown to be involved in RNA polymerase II assembly.
      • If not, then, in this sentence, the author should switch the positions of RPB5 and WDR92
    • Line 68 – “factors” should be “adaptors”
    • Line 68 – Why “so-called”?
    • Line 69 – “help bringing” should be “help bring”
    • Line 69 – “help bringing a specific set of clients, the PIKK ..” should be “help bring the PIKK family of kinases to R2TP”
    • Line 67 – 75: This whole paragraph is problematic
      • They do not cite Horejsi 2010 or 2014, which both show that PIH1D1 interactions with phosphorylated Tel2 are essential for mTOR stability.
        • He implies that this work has not been done yet
      • Why is the PAQosome review (Houry, 2018) being cited here?
    • Line 78 – “The PAQosome includes R2TP and R2TP-like complexes”
      • No, it does not.
    • Line 78 – “URI1 Prefoldin-like module” should be “URI1 Prefoldin-like complex” to keep consistency
    • Line 79 – “and all other proteins and adaptor required”
      • No! The PAQosome does not include the adaptors.

Section 2

  • Line 89 – Reference should be the R2TP cryo-EM structure, not their review.
  • Line 90 – “essential component not only of R2TP but also” should be “essential component of not only R2TP, but also several other”
  • Line 92 – Should say “serves as”
  • Line 95 – Should be “species: yeast,”
  • Line 99 – Should be “hexamer, and it is”
  • Line 101 – Should be “ … fold domain. Clients and adaptors ..”
  • Line 104 – Should be “(Figure 2B, C)’
  • Line 106 – Should be “suggests”

Section 3

  • Line 121 – Should be “predicted as disordered”
  • Line 122 – Should be “constituents of yeast R2TP,”
  • Lines 134-7 – I have no idea what they’re trying to say here.
  • Line 138 – Should be “significant”
  • Line 152 – The comma after “[24]” should be deleted

Section 4

  • Line 155 – “One of the most noticeable differences between yeast and human R2TP complexes is …” Should be “The most noticeable difference between yeast and human R2TP complexes is …”
    • The former would make “differences” the subject of the sentence and would therefore need a plural verb (ie., “are” instead of “is”)
  • Line 157 – Delete comma after “homolog”
  • Line 162 – Delete “each”

Section 5

  • Line 200 – “comples” should be “complex”
  • Line 202 – Should be “(Figure 2B, C)”
    • The dash should be used only when there are more than two panels

Section 6

  • Line 222 – Should be “isoform 2 being”
  • Line 232 – “In vitro” should be italicized
  • Line 239 – “Primary” should not be capitalized.
  • Line 241 – Should be “A known mutation found in PCD ...”
  • Line 244 – Should be “have roughly 25% sequence …”
  • Lines 246-50 – A run-on sentence that repeats itself.
  • Line 271 – Insert a comma after “detail”
  • Line 272 – Delete the comma after “functions”
  • Line 272 – “and/or specificity towards some clients” should be “and specificity towards some clients.”

Section 7

  • Line 275 – “require the action of the R2TP/URI1 Prefoldin based co-chaperone for their assembly, activation or maturation”
    • I don’t think it should be assumed that the URI1 prefoldin complex has an essential role in the aforementioned complexes
  • Line 277 – What are these “factors” they keep mentioning? They should just say “proteins” or “adaptors.”
  • Lines 281-2 – Should be, “The interaction of the RUVBL1-RUVBL2 ATPases with an RPAP3 C-terminal-like domain is the only element …”
  • Line 284-6 – Another awkward sentence that should be rephrased

Reviewer 2 Report

This is a nice, short review on what is known, in structural terms, of a very interesting complex, the R2TP complex. It should be accepted as it is, with a few minor changes (see below):

- I would add two more keywords: TPR domain and URI 1 Prefoldin-like complex

- Line 26 – The PAQosome is included in the title of part 1 without being defined. I would remove it from there

- Legend to Fig. 2. Since the model in Fig. 2B is based on the cryoEM averages shown in Fig. 2C, the order should be switched.

- Domains DI and DIII are not described in Fig. 2

- Line 199. “complex” instead of “comples”

- There is a comprehensive description of the role of the C-terminal part of RPAP3. However, although mentioned, there is not much information, certainly not structural, on the role of the TPR domains in the complex, which are critical in the co-chaperone role.

- It is slightly confusing the fact that the ATPase face of RUVBL1/2 is in the bottom in Figs. 1 and 3, and in the top in Fig. 4 (i.e. the latter is upside down with regard to the former). Perhaps this should be fixed.

Round 2

Reviewer 1 Report

Minor Revisions:

Line 43 – comma after “and”

Line 48 – “also” should go before “participates”

Line 93 – comma after “and”

Lines 121 – 124 – need to also mention that RPB5 and WDR92 are part of the PAQosome.

Lines 175-9 – “In 2017, two cryo-EM structure of yeast R2TP (Rvb1p-Rvb2p-Tah1p-Pih1p) reconstructed at medium resolution revealed that Rvb1p-Rvb2p formed a hexameric ring that scaffolds a Tah1p in complex with Pih1p.”

Line 179 – “Pih1p and Tah1p were located at”

Line 182 – “Significant” not “significance”

Line 199 – Omit “Color”

Line 210 – “PIH1D1 and most of RPAP3 are located in”

Line 212 – “and this allows R2TP to recruit”

Line 342 – “two isoforms, with isoform 2 being shorter as it lacks few residues”

Line 367 – Comma after “[3]”
